# Assessing risks of dengue, chikungunya and Zika transmission associated to *Aedes albopictus* in Chania, Greece, 2017–2018

Sk Shahid Nadim[1,2,‡], Francesco Menegale[1‡], Mattia Manica[1], Alexander R. Kaye[3,4], Georgios Balatsos[5], Marina Bisia[5], Verena Pichler[6], Piero Poletti[1], Stefano Merler[1], Alessandra della Torre[7], Robin N. Thompson[8], Antonios Michaelakis[5], Giorgio Guzzetta[1*]

**1** Centre for Health Emergencies, Bruno Kessler Foundation, Trento, Italy, **2** Department of Mathematics, SRM University-AP, Amaravati, Andhra Pradesh, India, **3** Mathematics Institute, University of Warwick, Coventry, United Kingdom, **4** Zeeman Institute for Systems Biology and Infectious Disease Epidemiology Research (SBIDER), University of Warwick, Coventry, United Kingdom, **5** Laboratory of Insects & Parasites of Medical Importance, Benaki Phytopathological Institute, Athens, Greece, **6** Saint Camillus International University of Health and Medical Sciences, Rome, Italy, **7** Department of Public Health and Infectious Diseases, "Sapienza" University, Rome, Italy, **8** Mathematical Institute, University of Oxford, Oxford, United Kingdom

‡ These authors share first authorship on this work.
* guzzetta@fbk.eu

## Abstract

The stable presence of the *Aedes albopictus* mosquito in Europe has set the stage for the emergence of tropical arboviral outbreaks (such as dengue and chikungunya), following the importation of infection by international travelers. Here, we leverage *Ae. albopictus* capture data collected weekly in Chania, Greece, in 2017 and 2018, to calibrate a model for assessing the potential epidemiological risks of mosquito-borne outbreaks such as dengue, chikungunya, and Zika. We estimated a peak density of female mosquitoes of 459 (95% Credible Interval, CrI: 424–508) per hectare in 2017 and 757 (95% CrI: 728–785) in 2018. The peak reproduction numbers occurred in early September and exceeded the epidemic threshold of 1 in 20–26% of the municipality area for dengue and in 40–70% for chikungunya (depending on the year). In contrast, we found a negligible risk of Zika transmission. We assessed the quantitative risks of outbreaks for both dengue and chikungunya, using two alternative measures, the Instantaneous Epidemic Risk (IER), and the Threshold Epidemic Risk (TER). We assessed quantitative differences in the two metrics and their determinants, showing that the IER tends to underestimate the risk of onward transmission early in the summer and to overestimate it in the second half of the season. This study identifies non-negligible risks of arboviral outbreaks in a country that, to date, has not recorded autochthonous transmission. It also underscores the importance of considering and adjusting for potential biases in traditional measures of epidemic risk.

**Data availability statement:** All relevant data are in the manuscript and its supporting information files.

**Funding:** This work was supported by EU funding within the NextGeneration EU-MUR PNRR Extended Partnership Initiative on Emerging Infectious Diseases (project no. PE00000007, INF-ACT) received by SM and AdT. This work was also supported by Wellcome through a Digital Technology Development Award (Climate-Sensitive Infectious Disease Modelling) granted to RNT, grant number 226057/Z/22/Z. The funders had no role in study design, data collection and analysis, decision to publish, or preparation of the manuscript.

**Competing interests:** The authors have declared that no competing interests exist.

## Author summary

*Aedes albopictus* is a highly invasive mosquito capable of transmitting arboviruses such as dengue, chikungunya, and Zika. In this work, we leverage mosquito capture data collected over two years of entomological surveillance in Chania - a major tourist town on the Greek island of Crete - to assess the probability of arboviral transmission of public health relevance following a potential introduction of the virus by an infected traveler. We found noteworthy risks of dengue and chikungunya outbreaks in the area during the summer and early fall, while the probability of Zika outbreaks was negligible. We also show that classical methods may underestimate the risk of arboviral outbreaks for viral introductions occurring in the early summer and overestimate them in the early fall. Our findings underscore the usefulness of entomological surveillance to assess arboviral risks in a region without previous overt transmission and provide methodological insights for performing such assessments.

## Introduction

*Aedes albopictus*, also known as the Asian tiger mosquito, is one of the most successful invasive mosquito species worldwide [1]. Initially native to tropical and subtropical regions of Southeast Asia, it has expanded its range dramatically in recent decades, particularly due to human activities such as global trade [2]. *Ae. albopictus*, first reported in Europe in 1979 in Albania [3], arrived through the importation of goods and has since spread in several countries, including Italy (first reported in 1990) [4] and Greece (in 2005) [5]. This mosquito vector has now established populations across much of southern and central Europe, displaying an ability to adapt to temperate climates [6]. This has been highlighted by several epidemiological and entomological surveillance studies focused on its distribution and abundance, conducted in Switzerland [7], Spain [8], Germany [9], Italy [10], and Greece [11]. *Ae. albopictus* is the vector of at least 23 arboviruses, including chikungunya, dengue and Zika [12], for which viral transmissibility generally increases with the mosquito population density [13].

With the species' continuing expansion in Europe, the risk of mosquito-borne diseases has increased, and a growing number of arboviral outbreaks have been observed in mainland Europe in recent years [14,15], including substantial case counts. Only in 2024, local dengue transmission totaled 83 cases in France and 213 cases in Italy; in 2025, as of October 1, 637 locally transmitted cases of chikungunya were recorded in France and 323 in Italy [14,16]. Chania, one of the most popular tourist destinations in Greece, presents a compelling setting for studying arboviral diseases such as dengue and chikungunya. *Ae. albopictus* was first detected in the area in 2014 [17]. A previous study found a moderately high climatic suitability in the region for the establishment and proliferation of the mosquito, enabling potential arboviral transmission following the importation of an infectious case from

endemic areas [18]. Furthermore, Chania International Airport has experienced significant growth in passenger traffic in recent years, rising from 3.29 million passengers in 2022 to over 3.88 million passengers between January and November 2024 [19,20]. This substantial increase in the number of travelers elevates the risk of introducing arboviral diseases from endemic regions, underscoring the importance of vigilant public health monitoring and vector control measures in the area. These factors, combined with its Mediterranean climate, make Chania a critical location for assessing epidemic seasonality and developing targeted vector control strategies [21].

This study aims to assess and quantify potential epidemiological risks associated with dengue, chikungunya, and Zika transmission in Chania. We used mathematical modeling techniques to estimate the population density of *Ae. albopictus* mosquitoes, determine basic reproduction numbers for the three viruses, and assess outbreak probabilities, i.e., the risk that cases introduced (e.g., via travelling from endemic areas) will lead to sustained local transmission, as opposed to the virus fading out with few or no secondary infections.

## Methods

### Study area

The study was conducted within the Municipality of Chania, located on the northwest coast of the Mediterranean island of Crete, Greece (35.48°N, 24.02°E), with a population of approximately 108,000 residents. An entomological surveillance network was established to monitor mosquito presence from May 1 to December 31, 2017, and from April 1 to December 31, 2018, in Akrotiri, an area spanning about 13 km² (35°33′N 24°08′E) encompassing rural and agricultural zones as well as Chania Airport.

The region has a hot-summer Mediterranean climate (Csa) with mean daily temperatures of 21.62°C in 2017 and 21.72°C in 2018 during the May – December period. Total rainfall for the surveillance months was 194 mm in 2017 and 462 mm in 2018. The Municipality of Chania has no recorded history of locally transmitted vector-borne diseases (VBDs), including those associated with *Aedes* mosquitoes.

### Entomological surveillance

The surveillance network was established in a rural area in the vicinity of Chania airport, comprising eight BG-Sentinel 2 traps for monitoring adult mosquitoes. Traps were established on private properties but exclusively in publicly accessible locations. The precise coordinates of each BG-Sentinel 2 trap were recorded using a global positioning system (GPS) device and trap locations remained fixed throughout the surveillance period [12].

Weekly samplings of BG sentinel traps were conducted according to standard operational procedures [4,22]. Nets from BG sentinel 2 traps were transferred from the field to the Laboratory of Insects and Parasites of Medical Importance at the Benaki Phytopathological Institute (BPI). Adults from BG-Sentinel traps were identified to the species level using standard morphological mosquito identification keys [23–25].

### Meteorological data

Meteorological data corresponding to the surveillance period of this study were provided by the Institute for Environmental Research of the National Observatory of Athens (IERSD/NOA). Meteorological data were obtained from a Davis-type automatic station, which continuously transmitted real-time measurements of pressure, temperature, humidity, precipitation, wind direction, and wind speed. This station, situated in the Akrotiri area near Chania Airport, was positioned at 35.53337° N, 24.06835° E, and an altitude of 137 meters [26].

### Human population data

Human population data for the residential areas of the Municipality of Chania were analyzed using a disaggregation approach based on 147 cells with at least 10 inhabitants per hectare, at a spatial resolution of 250 m x 250 m, for a total

covered surface of 918.75 hectares (9.19 km²). Data on human density were obtained from the Global Human Settlement Layer (GHSL) project [27].

## Population model and calibration

We used a mathematical model describing the population dynamics of *Ae. albopictus* across its entire life cycle, encompassing eggs, larvae, pupae, and adult females [28]. The model incorporates previously established estimates of temperature-dependent mortality rates, the egg deposition rate and the transition rates between developmental stages (i.e., from eggs to larvae, from larvae to pupae, and from pupae to adult mosquitoes) [29]. Free model parameters were the year-specific larval overcrowding [30] and the effectiveness of traps in capturing adult females [28,31], and were calibrated against adult female mosquito capture data throughout the season, using a Markov Chain Monte Carlo (MCMC) approach. We started with uniform prior distributions and employed random-walk Metropolis-Hastings sampling for 50,000 iterations to obtain the posterior distribution of parameters with which we estimated the population density of adult female mosquitoes. Model equations, parameter values and calibration details are reported in S1 Text.

## Risk of arboviral outbreaks

To estimate the risk of chikungunya, dengue and Zika outbreaks in Chania, we computed the basic reproduction numbers $R_0(t)$, which vary over time due to seasonal changes in vector abundance and in temperature-dependent parameters. We used analytical results from the classical host-vector SEIR-SEI model [29]:

$$R_0(t) = \beta^2 \phi^2 \frac{N_V}{N_H} \frac{\chi_V \chi_H}{\mu_V \gamma} \frac{\omega_V}{\omega_V + \mu_V}$$

where $\beta$ is the mosquito biting rate, $\phi$ is the proportion of blood meals taken from humans by mosquitoes, $N_H$ and $N_V$ are respectively the number of humans and mosquitoes, $\chi_V$ is the probability of transmission from an infected human to a susceptible mosquito per bite, $\chi_H$ is the probability of transmission from an infected mosquito to a susceptible human per bite, $\frac{1}{\gamma}$ is the human infectious period, $\omega_V$ is the mosquito extrinsic incubation rate, and $\mu_V$ is the mosquito mortality rate (for ease of notation, we removed the time and temperature dependence of the parameter values in the right-hand side of the equation). For the mosquito population, $N_V$, we used estimates associated with the trap with highest estimated abundance over each year. A full description of the model parameters is provided in Table D in S1 Text. We drew 1,000 samples from the distributional estimates of literature-estimated parameters and from the posterior distribution of the mosquito density calculated from the population model. Temperature-dependent epidemiological parameters were used for dengue, while for chikungunya and Zika only temperature-independent estimates were available. Temperature-dependent parameters for chikungunya, adapted from dengue, were also explored in a sensitivity analysis (see S1 Text). We computed $R_0(t)$ separately for each of the 147 cells of the residential area of Chania using the corresponding human population density (i.e., spatial correlation across cells was neglected). Even when the basic reproduction number, $R_0$, is greater than one, the stochastic process of infection after a single infectious individual is introduced into a population may result in the fade out of transmission chains [32,33]. Under the implicit assumption that both the number of vectors and the epidemiological parameters remain constant from the time of introduction onwards, the probability that an initial infection will lead to an outbreak of public health relevance can be explicitly quantified in closed form [34]. Following [35], we term this estimate the "Instantaneous Epidemic Risk" (IER, expressed as a percentage; see S1 Text) and we computed it over time for each separate cell of the municipality. At any day $t$ of the season, the IER is given by:

$$IER(t) = 1 - \frac{R_{VH}(t) + 1}{R_{VH}(t)(R_{HV}(t) + 1)}$$

where

$$R_{VH}(t) = (\beta \phi \chi_H)/\mu V$$

and

$$R_{VH}(t) = \chi_V \beta \phi (N_V \omega_V)/(\gamma N_H (\omega_V + \mu_V))$$

We then summarized these results by estimating the epidemic season length (calculated as the number of days with an IER exceeding zero) and the average IER across the epidemic season.

Due to its assumption of constant transmissibility after the virus invades the population, the IER is an approximation of the actual risk of outbreaks. A more precise estimation can be evaluated by computational simulations where a single infectious individual is introduced in the population and mosquito abundances and epidemiological parameters are updated over time [31]. For each cell in the municipality, we ran 100,000 simulations of a stochastic model analogous to the SEIR-SEI model, using an adaption of the Gillespie algorithm for systems with temporally varying parameters (see S1 Text). The 100,000 simulations were made up of 100 simulations for each of the 1,000 samples from the distribution of literature-estimated parameters and from the posterior distributions of the mosquito density time-series. We then determined the risk of an outbreak of public health relevance for each cell by counting the proportion of the 100,000 simulations that resulted in 10 infections or more. This estimate of the risk is termed "Threshold Epidemic Risk" (TER, also expressed in percentage) [31]. Due to the high computational burden, we evaluated this quantity for virus introductions occurring on the first day of each month between June and October, for both 2017 and 2018. Full details of the computational methodology for computing the TER are reported in S1 Text.

To assess the drivers of differences between estimated values of the IER and TER, we fit a linear regression model. We considered as the dependent variable the difference between the TER and IER at the dates of introduction for which the TER was calculated, while independent variables were the date of introduction of the first infectious case and the cell's host population density. The date of introduction was considered as a categorical variable (since only 5 dates per year were evaluated for the TER), with August 1st being considered as the reference term (intercept) in the regression model. We only considered cells for which both the TER and IER were non-zero, to avoid distortion effects introduced by the positivity constraint on the TER and IER. The S1 Text reports full details of the regression model.

## Results

The entomological surveillance collected a total of 730 adult females of *Ae. albopictus* across the eight traps between May 1 and December 31, 2017, and 735 from April 1 to December 31, 2018. The model reproduced the general seasonal trends in the average mosquito captures (Fig 1A and 1B), as well as capture data from the individual traps (see S1 Text). The trap with the highest abundance collected a total of 268 adult females in 2017 and 453 in 2018, with peaks of respectively 34 and 58 captures in August. The resulting estimated mosquito abundance shows substantial variations in mosquito densities between the two seasons, with a mean peak of 459 (95% Credible Interval, CrI: 424–508) females per hectare in 2017 and 757 (95% CrI: 728–785) in 2018 (Fig 1C and 1D), occurring in both years during the second half of September.

To illustrate seasonal temporal trends in $R_0$ for dengue, chikungunya and Zika in 2017 and 2018, we simulated an illustrative cell with an assumed human population density of 40 inhabitants/ha. Peak $R_0$ values for both diseases were noted in mid-September 2017 and early September 2018, respectively (Fig 2). Similar $R_0(t)$ estimates for chikungunya were obtained in a sensitivity analysis in which temperature-dependent parameters were used (see S1 Text).

Fig 3 shows the distribution of peak seasonal values of $R_0(t)$ for dengue, chikungunya and Zika, associated with the variability in human population density across the 147 cells. The peak $R_0$ for dengue in 2017 was above the epidemic

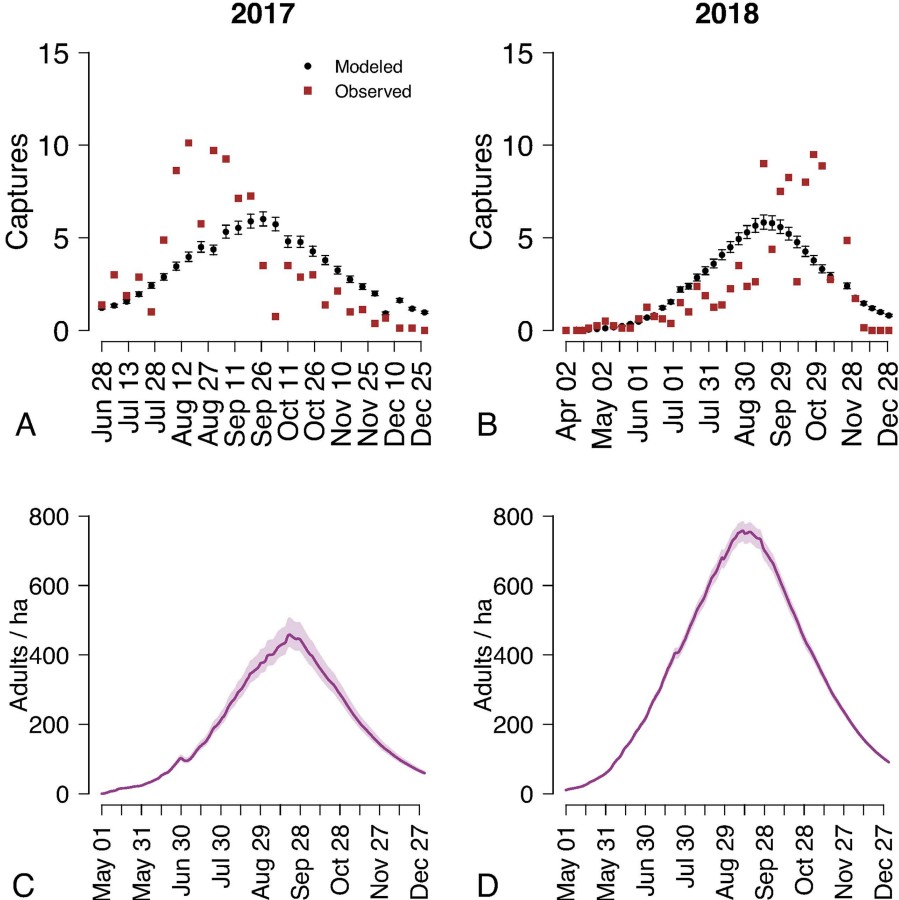

**Fig 1. Captures of female *Aedes Albopictus* and estimated abundance in Chania, Greece, 2017–2018.** A) Average adult female mosquito captures (red squares) per weekly trapping session in 2017, and corresponding model estimates (black dots: mean; whiskers: 95% CrI). B) As A), but for 2018. C) Model estimates of daily female adult mosquito density in the trap with highest estimated abundance in 2017. Solid lines: mean; shaded areas: 95% CrI. D) As C), but for 2018.

threshold of 1 in 20% of cells, with the highest value at 2.8. In 2018 the higher mosquito density resulted in 26% of the cells having a peak $R_0$ for dengue above 1, with the highest value at 3.9. A similar trend, with even more pronounced changes across the two years, was observed for chikungunya: about 40% of the cells in 2017 and 70% in 2018 had peak $R_0$ values greater than 1, and the maximum value increased from 5.1 to 8.1. Overall, at least one third of the residential area of the municipality was exposed to a reproduction number greater than one for both diseases at some point in each year. In contrast, we found a negligible risk of Zika transmission, with $R_0$ remaining systematically below one in all cells throughout 2017 and 2018.

Fig 4 shows the distribution of epidemic season lengths (representing the number of days with a non-zero IER following the importation of an infectious case) for dengue and chikungunya, and the average IER over the season. The epidemic seasons of dengue lasted more than two months in less than 3% of cells during 2017, and in over 12% during 2018. The average IER exceeded 20% in less than 3% of cells during 2017 and in over 10% during 2018. Longer season lengths and higher average IER were associated with chikungunya, due to the higher competence of *Ae. albopictus* for its transmission [36]: during 2017, the epidemic season was estimated to be longer than two months in over 20% of cells, with about 15% of cells having an average IER higher than 20%; during 2018, the season length exceeded two months in

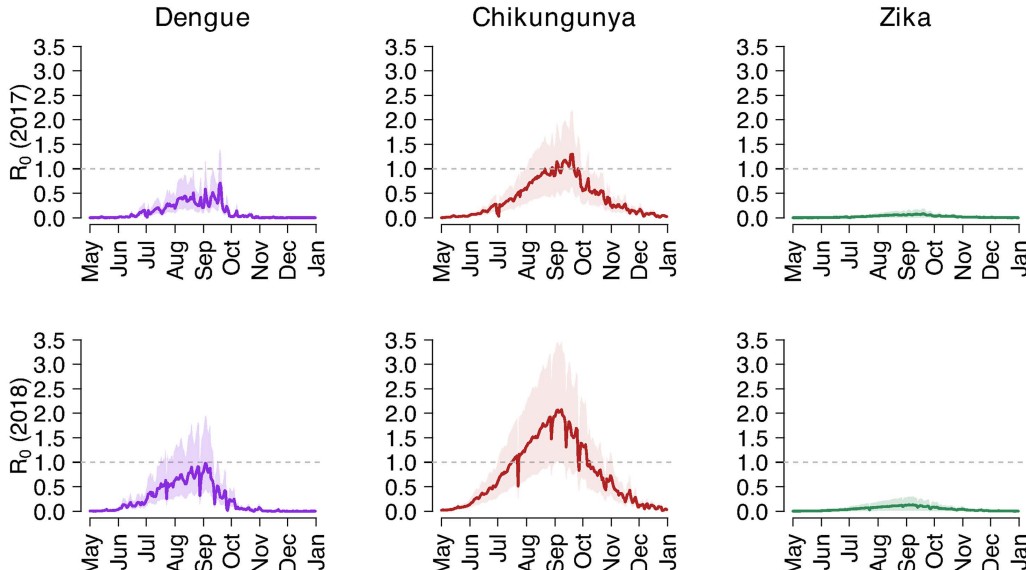

**Fig 2. Illustrative model-estimated temporal trends of the basic reproduction numbers of dengue, chikungunya and Zika in Chania, Greece, over 2017 and 2018, assuming a human population density of 40 individuals/ha.** Solid lines: mean; shaded areas: 95% CrI.

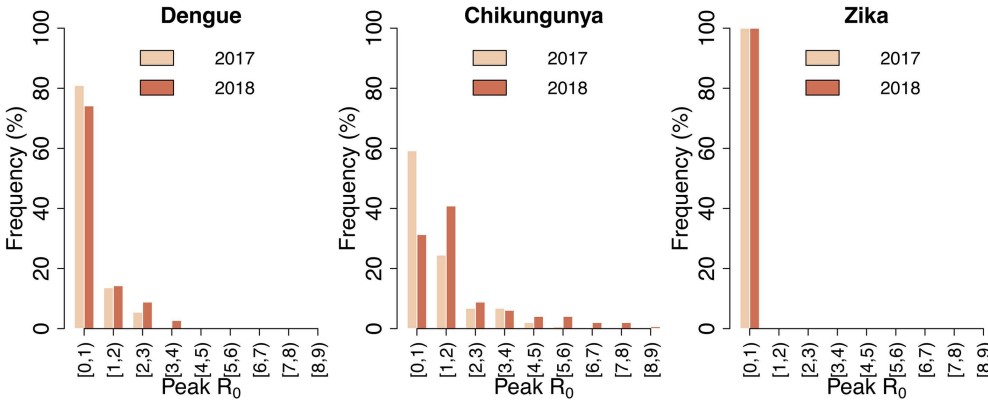

**Fig 3. Distribution of peak estimated $R_0$ values for dengue, chikungunya, and Zika across 147 cells constituting the municipality of Chania, Greece, in 2017 and 2018.**

almost 40% of cells (with peaks of over 5 months), and 30% were associated with an average IER higher than 20% (with peaks of more than 60%).

Fig 5 shows the spatial distribution of both the TER and IER for dengue and chikungunya in Chania, assuming the introduction of an infectious human case in each of the 147 cells of the municipality on September 1 in 2017 and 2018, close to the seasonal peak in mosquito abundance. Both measures indicate a higher risk for northern (coastal) and southern (peripheric) cells, as a consequence of the lower human population density compared to the central neighborhoods; in this case, the TER tended to estimate lower epidemic risks than the IER, reflecting the reduction in transmissibility following the peak (as noted in the Methods, unlike the TER, the IER assumes constant transmissibility following virus introduction).

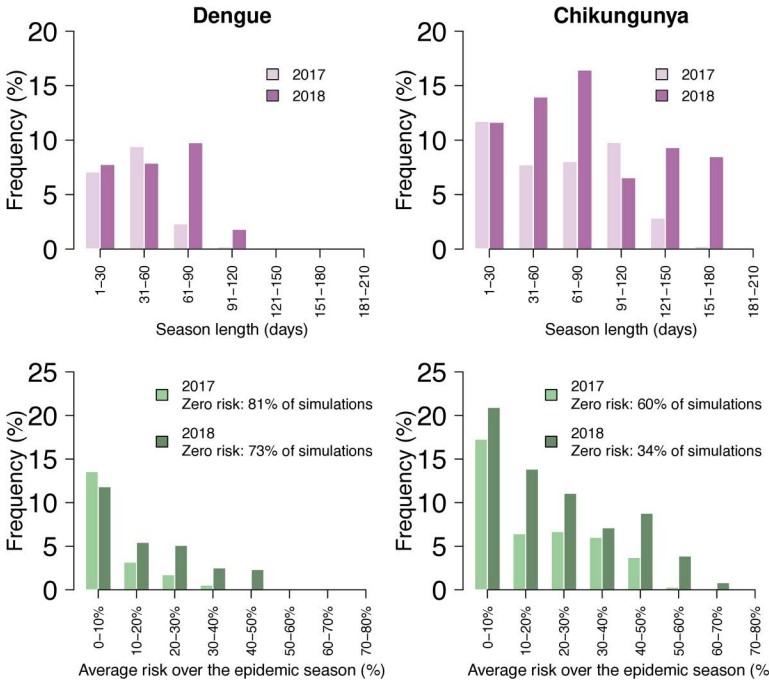

**Fig 4. Distribution of the epidemic season length and average risk over the epidemic season, measured through the IER, for dengue and chikungunya in Chania, Greece, in 2017 and 2018.**

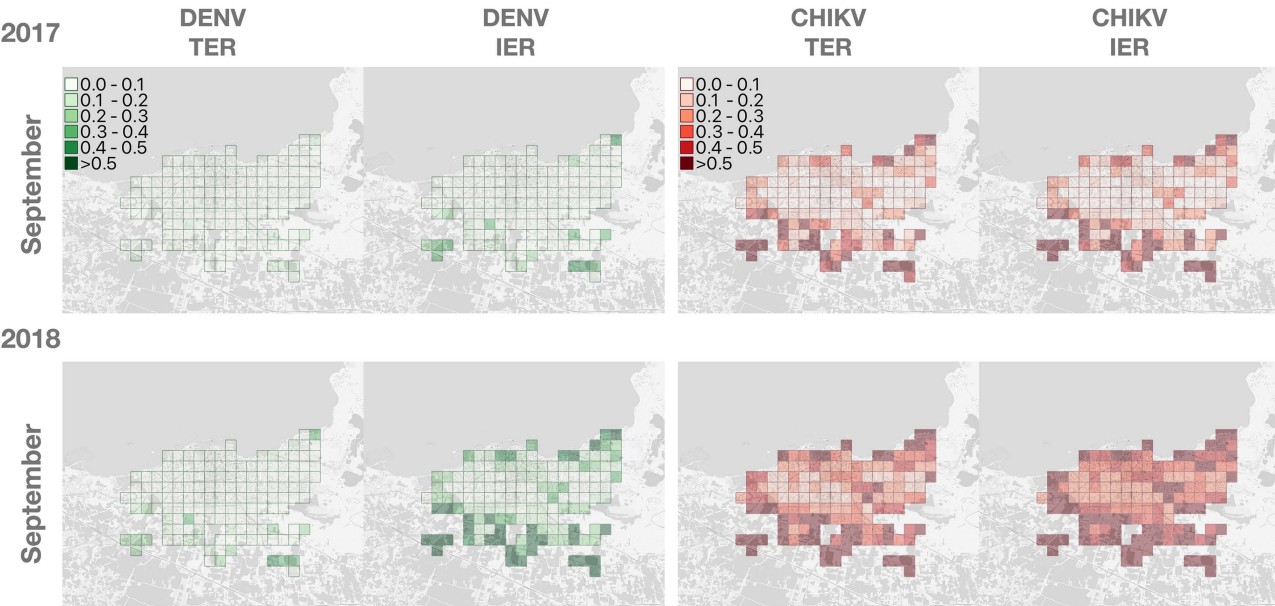

**Fig 5. Estimated Threshold Epidemic Risk (TER) and Instantaneous Epidemic Risk (IER) for dengue and chikungunya for introductions of a case on September 1, 2017 and 2018.** Estimates are provided at a spatial scale of 250 m × 250 m, focusing on 147 cells of the municipality of Chania with a human density higher than 10 individuals per hectare. Maps were created using QGIS software version 3.30.2. Background map layer was obtained from OpenStreetMap (https://www.openstreetmap.org) and is made available under the Open Database License (http://opendatacommons.org/licenses/odbl/1.0/). Any rights in individual contents of the database are licensed under the Database Contents License (http://opendatacommons.org/licenses/dbcl/1.0/).

The differences between the values of the TER and IER metrics are displayed in Fig 6. The figure shows that the TER and the IER had a similar temporal trend and magnitude for chikungunya, with peak values for introductions in early September; however, there was a more substantial contrast between the IER and the TER for dengue. For both dengue and chikungunya, the IER tended to underestimate the probability of outbreak, compared to the TER, in the early part of the mosquito season (June and July) and to overestimate in the later part (September and October), due to temporal variations in transmissibility following the introduction of the virus. The relationship between the TER and IER estimates was assessed using a regression model where we express the difference between the TER and IER in terms of its precise value (as a difference between probabilities, each between zero and one) rather than as a

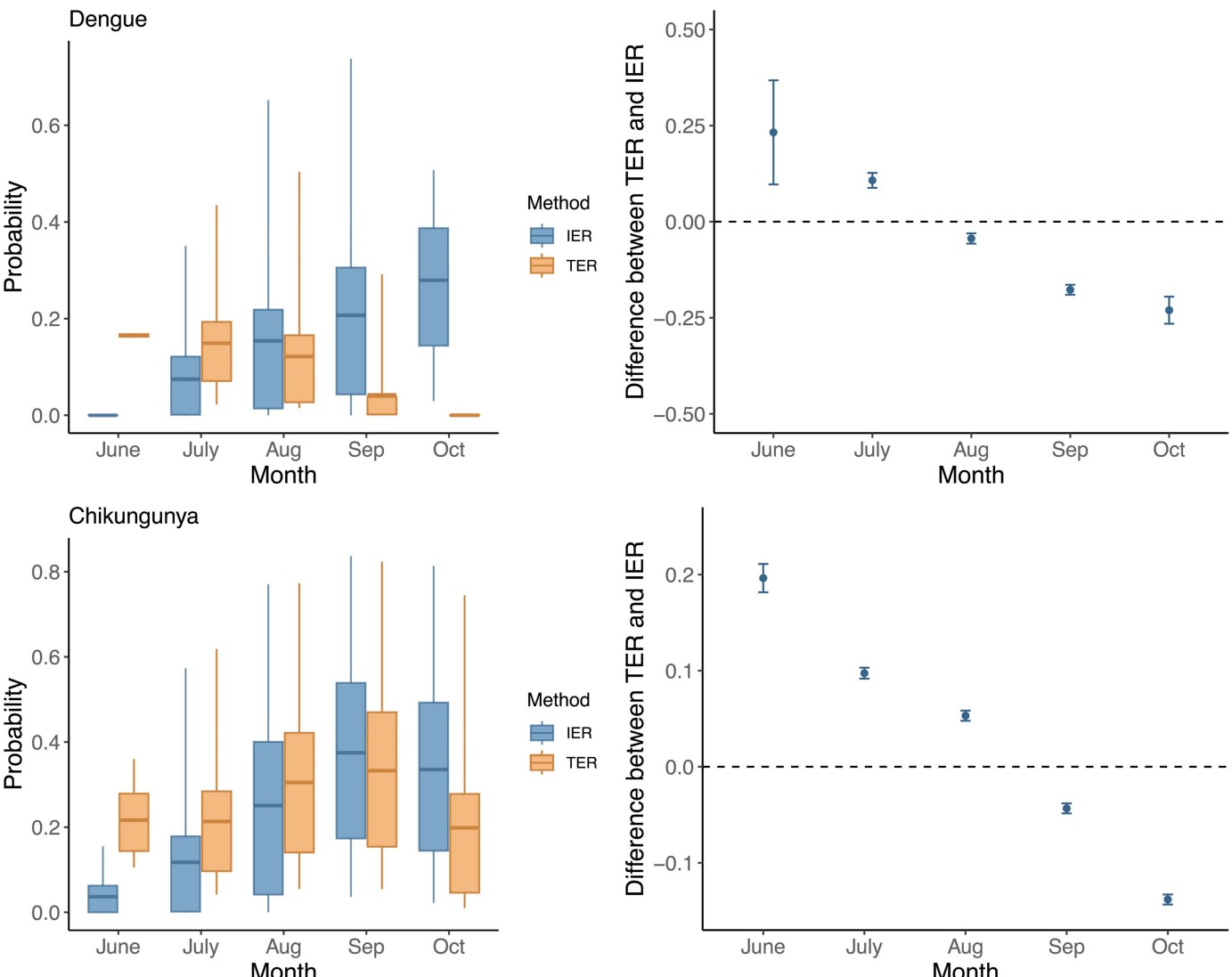

**Fig 6. Relationship between TER and IER estimates of the probability of dengue and chikungunya outbreaks, assuming the introduction of a viremic index case on the first day of June, July, August, September and October of 2017 and 2018.** Left: distributions of the probability of outbreaks, obtained by pooling together estimates from both years across cells for which both TER and IER values were non-zero; horizontal bar: mean; boxes: interquartile range; vertical lines: 95 percentile range. Right: coefficients of a linear regression model for the difference between TER and IER. Points represent the mean difference, and the vertical bands represent the 95% confidence interval.

percentage, to avoid the risk of misinterpretation as a relative difference. Results show that there was a markedly linear trend in coefficients over time of introduction of the index case. Higher human population densities tended to reduce the IER more than the corresponding TER estimates, with a stronger effect for dengue than for chikungunya: in particular, changes in 1 unit of host density corresponded to increases of 0.05 in the TER-IER difference for dengue and 0.01 for chikungunya.

## Discussion

We estimated the basic reproduction numbers and probability of dengue, chikungunya and Zika outbreaks following the potential introduction of a viraemic index case in the municipality of Chania, Greece. To calculate the probability of an outbreak, we used two alternative approaches: the classical derivation from theoretical SEIR-SEI stochastic models [34], termed the "infection epidemic risk", IER [35], and a simulation-based estimate that considers a threshold in the outbreak size, called the "threshold epidemic risk", TER [30]. To inform our analyses, we leveraged weekly mosquito capture data collected throughout the 2017 and 2018 seasons to calibrate a population dynamics model and estimate the abundance of adult female *Ae. albopictus* over time.

We found a non-negligible risk of local dengue and chikungunya transmission in both years, with marked temporal and geographical heterogeneity. We estimated a 65% increase in the peak mosquito population in 2018 compared to 2017, which resulted in substantially increased reproduction numbers, and therefore in larger areas of the municipality exposed to higher outbreak risks and for a longer time; in 2018, up to 70% of the municipality exceeded the epidemic threshold for chikungunya. As previously found for temperate areas in Italy [37], the reproduction number of Zika in Chania remained at all times below the epidemic threshold of one over the two considered years, independently of the location of the imported case. Similarly to results previously reported for Northern Italy [28], chikungunya tended to have longer epidemic seasons and higher risks of outbreaks compared to dengue, due to the generally higher competence for chikungunya transmission [36]. This result was robust to using dengue-inspired temperature-dependent functional forms for epidemiological parameters of chikungunya rather than constant values from experimental studies (see S1 Text). There was a marked heterogeneity also with respect to the specific area of the municipality where the index case was assumed to spend their time during their infectious period: coastal and peripheral areas of the town, being characterized by lower human densities (and therefore a higher probability of each host being bitten by mosquitoes), had higher risks compared to the more populated central neighborhoods, again in agreement with previous results from the Lazio region in Central Italy [38].

The quantitative estimation of the probability of outbreaks depended on the considered measure. While the classical IER estimate can be analytically computed for introductions occurring at any time of the year, it relies on the assumption that transmissibility conditions (mosquito abundance and temperature-dependent transmission parameters) do not change in the early phase of the outbreak. However, because the generation time of arboviral infections is of the order of two weeks [39,40], and the mosquito abundance may change substantially over this time frame, this assumption might be inaccurate, leading to incorrect risk estimates, particularly at the start and the end of the mosquito season (which are characterized by rapid variations in the vector density). On the other hand, the TER provides a robust simulation-based measure that incorporates temporal changes in transmissibility conditions and therefore is more accurate, yet is also more computationally intensive to calculate; therefore, we only computed it for selected days of introduction of the first viremic case, i.e., the first day of each month from June to October for both 2017 and 2018. By comparing IER and TER estimates via a linear regression model, we found that the IER tended to underestimate the TER in the first half of the season and to overestimate it in the second half, and was more sensitive to variations in human population density. The TER is able to account for changes in transmission conditions after the viral introduction, so it is larger than the IER when favourable conditions are coming and smaller when worse conditions are coming [41]. While the IER was relatively in agreement with TER in the estimation of the temporal trend of the outbreak risk for chikungunya, it was markedly different in its estimates

for dengue probably due to the inclusion of temperature-dependent epidemiological parameters. This finding emphasizes the need to carefully evaluate the limitations of the IER when estimating epidemic risks.

A limitation of this study was the reliance on mosquito capture data from traps located in a rural area near the Chania international airport. While this site is slightly outside of the residential area of the municipality, it serves as one of Greece's major points of entry, handling over 3 million passengers annually in 2017–2018 [20]. In particular, we used the maximum mosquito abundance estimated across traps for each year, which does not necessarily reflect the mosquito population within the residential area of the town. The choice of the maximum estimated mosquito abundance seemed the most appropriate given the available data for two reasons: first, urban areas are known to be more favorable for *Ae. albopictus* proliferation than rural areas; second, it is more risk averse to consider the worst observed conditions to avoid underestimating potential risks. However, this choice needs to be considered when interpreting our results. The study did not include possible spatial heterogeneities in microenvironmental conditions across the town, including the availability of food, shaded areas, breeding sites, and the abundance of predators, which can affect the development and survival rates across different mosquito developmental stages; similarly, the availability of temperature data from a single weather station for the whole municipality did not enable the explicit consideration of local microclimatic conditions that may influence mosquito proliferation and arboviral risks. Our TER estimates are sensitive, especially in the second half of the mosquito season, to the choice of the number of infections for an outbreak to be considered relevant for public health [30]. Our choice of 10 infections or more includes either outbreaks with high transmissibility, or, given a generation time of two to three weeks for arboviral outbreaks [39,40], stuttering transmission chains over a significant portion of the breeding season; we contend that either of these possibilities are worthy of public health authorities being alerted to an ongoing outbreak. The computation of the probability of outbreaks via both the IER and the TER is also subject to the assumption that the outbreak develops in the same geographic cell as the one in which the imported index case arrives, reflecting the limited mobility of infectious mosquitoes. However, in practice, human mobility may cause secondary cases to appear in other cells of the municipality, seeding new transmission foci and potentially altering the outbreak risk. We do not expect these effects to alter our general conclusions substantially; however, further research is needed to evaluate the quantitative impact of local human mobility on the probability of an outbreak in a heterogeneous landscape of transmissibility.

Another study limitation is the assumption of a constant human population density: in a highly touristic destination like Chania, the inflow of tourists during summer months can increase local population densities, thereby reducing the probability of onward transmission; on the other hand, if an outbreak is initiated, the presence of a large tourist population will increase the likelihood of seeding the infection in other locations where there is potential for arboviral transmission. The potential role of temporary tourist populations in arboviral transmission dynamics needs to be elucidated. In 2017, a chikungunya outbreak started around early June in Anzio [38] in the Lazio region of Italy, a popular seaside destination for inhabitants of the region; in addition to over 200 locally transmitted cases in Anzio, the outbreak seeded a number of smaller self-contained outbreaks in the surrounding areas, including the municipality of Rome (adding up to approximately 200 further cases) [40], and a secondary large outbreak of about 100 cases in the municipality of Guardavalle Marina, 600 km from Anzio.

We did not account for vector control measures in the area of mosquito collection that may have affected calibration of the model; however, we point out that the mosquito trapping area was a rural one, therefore we believe it is unlikely that it was subject to any intervention. Vector control measures may have been implemented within the urban area of Chania, thereby altering the outbreak risk. Because our simulations do not explicitly account for interventions, the estimates must be considered as potential risks in absence of interventions. Finally, the impact of human behavioral factors was not considered: protective actions such as the use of repellents, time spent outdoors and socio-economic characteristics (e.g., housing conditions) should also be investigated to improve the accuracy of risk assessments.

Based on the Greek National Public Health Organism, in recent years, only imported cases of these diseases have been recorded, in limited numbers, among travelers arriving from endemic countries abroad [42,43]. Laboratory experiments have also demonstrated the vector competence of *Aedes* populations in the region, even when females were

offered moderately low initial virus titers [44], suggesting that vector competence in natural field conditions could be higher, potentially amplifying transmission risks; on the other hand, field conditions may lead to higher mosquito mortality, thereby decreasing transmission risks. Additionally, climate change, through rising temperatures and altered precipitation patterns, is likely to increase mosquito populations and extend their seasonal activity, further elevating the risk of arboviral outbreaks [45–47]. These considerations highlight the necessity for entomological surveillance and the implementation of targeted vector control strategies to mitigate potential public health impacts related to invasive mosquito species [4,48,49].

This study suggests that the abundance of *Ae. albopictus* in Chania may be sufficient for the emergence of outbreaks of chikungunya and dengue in the future, especially in coastal and peripheral areas of the town with lower population density. This result obtained from entomological empirical data corroborates previous predictions obtained from global [50,51] and continental [31] models of arboviral transmission. In contrast, as previously found for temperate areas in Italy, the risk of Zika outbreaks was estimated to be negligible due to the poor competence of European *Ae. albopictus* strains [52]. However, this may change in the future following viral evolution and adaptation: it is therefore important to monitor changes in the competence of mosquito populations for different arboviruses over time. The existence of non-negligible arboviral outbreak risks suggests the importance of integrated vector management to reduce the risks of onward transmission, especially between mid-summer and early autumn. Previous studies demonstrated the cost-effectiveness of undertaking routine larvicide treatment in public spaces for medium-to-small municipalities [53], and the effectiveness of door-to-door interventions with public awareness campaigns for breeding site removal [54,55]. Enhancing surveillance systems to facilitate the early detection of local arboviral transmission can be critical to reduce the burden of potential ensuing outbreaks [40,56]. From a methodological point of view, this study also confirmed with a practical example that the classical, frequently used theoretical formulation for the computation of arboviral outbreak risks may be inaccurate in certain scenarios [30], underestimating risks in the first half of the season and overestimating them in the second half. Finally, the collection of mosquito abundance data at multiple sites and over time remains critical to assess and monitor the evolution of arboviral risks, in the context of rising temperatures and growing arboviral transmission in temperate areas [6]. Recent technological advances [57] may make entomological surveillance efforts less labor-intensive, facilitating the expanded assessment of arboviral risks [31].

## Supporting information

**S1 Text. Supplementary text containing full details about the model and its parametrization, as well as additional results.**
(DOCX)

**S1 Data. Mosquito capture and temperature data.**
(ZIP)

## Author contributions

**Conceptualization:** Alessandra della Torre, Robin N Thompson, Antonios Michaelakis, Giorgio Guzzetta.

**Data curation:** Shahid Nadim Sk, Francesco Menegale, Mattia Manica, Georgios Balatsos, Marina Bisia.

**Formal analysis:** Shahid Nadim Sk, Francesco Menegale, Mattia Manica, Alexander R Kaye.

**Funding acquisition:** Stefano Merler, Alessandra della Torre, Robin N Thompson.

**Investigation:** Shahid Nadim Sk, Francesco Menegale, Mattia Manica, Alexander R Kaye, Piero Poletti, Robin N Thompson, Giorgio Guzzetta.

**Methodology:** Shahid Nadim Sk, Francesco Menegale, Mattia Manica, Alexander R Kaye, Piero Poletti, Robin N Thompson, Giorgio Guzzetta.

**Software:** Shahid Nadim Sk, Francesco Menegale, Mattia Manica.

**Supervision:** Robin N Thompson, Giorgio Guzzetta.

**Visualization:** Shahid Nadim Sk, Francesco Menegale, Mattia Manica, Georgios Balatsos.

**Writing – original draft:** Shahid Nadim Sk, Francesco Menegale, Giorgio Guzzetta.

**Writing – review & editing:** Shahid Nadim Sk, Francesco Menegale, Mattia Manica, Alexander R Kaye, Georgios Balatsos, Marina Bisia, Verena Pichler, Piero Poletti, Stefano Merler, Alessandra della Torre, Robin N Thompson, Antonios Michaelakis, Giorgio Guzzetta.

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
