## [Decision Letter · Decision Letter 0]

17 Aug 2025

Assessing risks of dengue, chikungunya and Zika transmission associated to Aedes albopictus in Chania, Greece, 2017-2018

Dear Dr. Guzzetta,

Thank you for submitting your manuscript to PLOS Neglected Tropical Diseases. After careful consideration, we feel that it has merit but does not fully meet PLOS Neglected Tropical Diseases's publication criteria as it currently stands. Therefore, we invite you to submit a revised version of the manuscript that addresses the points raised during the review process.

Please submit your revised manuscript within 60 days Oct 16 2025 11:59PM. If you will need more time than this to complete your revisions, please reply to this message or contact the journal office at plosntds@plos.org. Please include the following items when submitting your revised manuscript:

We look forward to receiving your revised manuscript.

Kind regards,

Olaf Horstick, FFPH(UK)

Academic Editor

David Safronetz

Section Editor

Shaden Kamhawi

co-Editor-in-Chief

Paul Brindley

co-Editor-in-Chief

**Journal Requirements:**

At this stage, the following Authors/Authors require contributions: Shahid Nadim Sk, Francesco Menegale, Mattia Manica, Alexander R Kaye, Georgios Balatsos, Marina Bisia, Verena Pichler, Piero Poletti, Stefano Merler, Alessandra Della Torre, Robin N Thompson, Antonios Michaelakis, and Giorgio Guzzetta. Please ensure that the full contributions of each author are acknowledged in the "Add/Edit/Remove Authors" section of our submission form.

3) Please ensure that all Figure files have corresponding citations and legends within the manuscript. Currently, Figure 2 in your submission file inventory does not have an in-text citation. Please include the in-text citation of the figure.

4) Please ensure that the figures are labeled correctly in a numerical order in the manuscript.

5) We have noticed that you have uploaded Supporting Information files, but you have not included a list of legends. Please add a full list of legends for your Supporting Information files after the references list.

Potential Copyright Issues:

i) Figures 5, S2, S7, S8, S9, S11, and S12. Please (a) provide a direct link to the base layer of the map (i.e., the country or region border shape) and ensure this is also included in the figure legend; and (b) provide a link to the terms of use / license information for the base layer image or shapefile. We cannot publish proprietary or copyrighted maps (e.g. Google Maps, Mapquest) and the terms of use for your map base layer must be compatible with our CC BY 4.0 license.

7) Please amend your detailed Financial Disclosure statement. This is published with the article. It must therefore be completed in full sentences and contain the exact wording you wish to be published.

3) If any authors received a salary from any of your funders, please state which authors and which funders.

8) Thank you for stating "All the authors declare that they have no conflicts of interest." Please modify your Competing Interest statement on the submission form to the standard "The authors have declared that no competing interests exist."  

**Comments to the Authors:**

**Please note that one of the reviews is uploaded as an attachment.**

**Reviewers' Comments:**

Reviewer's Responses to Questions

**Key Review Criteria Required for Acceptance?**

**Methods**

-Are the objectives of the study clearly articulated with a clear testable hypothesis stated?

-Is the study design appropriate to address the stated objectives?

-Is the population clearly described and appropriate for the hypothesis being tested?

-Is the sample size sufficient to ensure adequate power to address the hypothesis being tested?

-Were correct statistical analysis used to support conclusions?

-Are there concerns about ethical or regulatory requirements being met?

Reviewer #1: Clarity of objectives and hypothesis: The primary objective – to assess arboviral transmission risks in Chania – is clearly articulated. An implicit hypothesis is that there is a non-negligible risk of local transmission for dengue, chikungunya, and/or Zika following an imported case. This could be stated more explicitly.

Parameterization: The model relies on temperature-dependent parameters. For chikungunya and Zika, temperature-independent epidemiological parameters were used. Given that chikungunya risk was found to be notably higher than dengue (whose model included temperature-dependent parameters), a sensitivity analysis on the key time-independent parameters for chikungunya is required to assess the robustness of this important finding and to understand how sensitive the higher chikungunya risk is to these fixed parameter values.

Definition of "Widespread Outbreak" in TER: The TER calculation defines an outbreak as ≥10 infections. While any threshold is somewhat arbitrary, the rationale for this specific value should be discussed and the authors should briefly consider how varying this threshold might impact TER estimates and the comparison with IER.

Reviewer #2: (No Response)

Reviewer #3: (No Response)

**Results**

-Does the analysis presented match the analysis plan?

-Are the results clearly and completely presented?

-Are the figures (Tables, Images) of sufficient quality for clarity?

Reviewer #1: Te Results section (particularly concerning Figure 3 and Figure 4) tends to be more descriptive of the figures' content rather than providing an immediate interpretation of what those descriptions signify. While the Discussion section later contextualizes these findings, weaving in some level of interpretation directly within the Results can indeed enhance clarity and demonstrate the added value of each figure more immediately

Figure 6: The purpose of the IER/TER comparison could be more clearly signposted. It appears to serve a dual role: 1) to demonstrate that differences exist between the metrics, and 2) to explore the determinants of these differences.

Perhaps structuring the paragraph in the Results more explicitly. For instance, you could first discuss the direct comparison of IER and TER (from the box plots), highlighting the general patterns. Then you could introduce the regression analysis specifically as a means to understand why these differences occur and what factors influence them.

Also please see below (data presentation modifications) for minor comments on figures.

Reviewer #2: The figures and results are clearly communicated, and the analysis plan matched the final results. However, no results of the raw capture data, which is an important part of the methodology is presented in the results.

Reviewer #3: (No Response)

**Conclusions**

-Are the conclusions supported by the data presented?

-Are the limitations of analysis clearly described?

-Do the authors discuss how these data can be helpful to advance our understanding of the topic under study?

-Is public health relevance addressed?

Reviewer #1: There are several additional limitations or areas for more detailed discussion:

- The authors should clarify whether any routine or reactive vector control measures were ongoing in Chania during the 2017-2018 study period. Such activities could significantly influence local mosquito densities and affect model calibration and risk assessments, acting as an unmeasured confounder.

- While the study used meteorological data, mosquito development and survival are highly influenced by microclimatic conditions (e.g., in shaded areas, breeding sites).

- human behavioral factors e.g., human mobility, time spent outdoors, use of repellents, or socio-economic factors influencing housing conditions

Reviewer #2: There is discussion of the results, and limitations of the methods used. There is also public health relevance mentioned, although a slight fleshing out of this could be beneficial.

Reviewer #3: (No Response)

**Editorial and Data Presentation Modifications?**

Reviewer #1: Introduction:

Line 61-62: "Surveillance studies have been conducted in..." Please clarify if these refer to mosquito surveillance, disease surveillance, or both. A brief elaboration on the nature of these cited surveillance studies (e.g., "focused on vector distribution and abundance") would add context.

Line 63-64: The statement "for which viral transmissibility is directly proportional to the mosquito population density" is a slight oversimplification. Consider rephrasing to something like "is strongly influenced by" or "generally increases with" mosquito population density.

The labeling of Figures in the main text captions appears incorrect from Figure 4 onwards in the submitted manuscript

Figure 5 : For panels showing generally very low risk (eg Dengue-TER) it would be helpful to include the boundary of the municipality of Chania as a light grey overlay or base map. This would provide better spatial context and make it easier to discern the assessed area, especially where many cells are blank or light-colored.

Figure 6 : The notations "Changes in 1 unit of host density corresponds to increase of X in the difference" directly on the right panels, while informative, contribute to visual clutter. Consider moving this quantitative detail to the figure caption or the main text

Reviewer #2: (No Response)

Reviewer #3: (No Response)

**Summary and General Comments**

Reviewer #1: This manuscript investigates the transmission risk of dengue, chikungunya, and Zika viruses by Aedes albopictus in Chania, Greece, a region with established vector presence but no history of local arboviral outbreaks. The study is timely, given the increasing establishment of Ae. albopictus in Europe and the potential for virus importation via international travel, particularly to tourist hotspots like Chania.

Strengths of the study include its application of a mathematical model calibrated with local, multi-year entomological surveillance data, its assessment of multiple arboviruses, and its insightful comparison of two different risk metrics (IER and TER). The finding of a non-negligible risk, particularly for chikungunya, and the spatial identification of higher-risk areas (peripheral, less densely populated zones) provide importnat information for public health authorities. The discussion on the limitations of the simpler IER metric is a valuable methodological contribution.

The primary weaknesses lie in the limitations imposed by some methodological assumptions that need further justification or exploration as outlined above.

Reviewer #2: This is a publication examining the risk of transmission of three arboviruses (dengue, Zika, and Chikungunya, in Chania, Greece, using mosquito surveillance, population, and climate data in a model to calculate the probability of local transmission. They found risks of outbreaks of chikungunya and dengue in this context and discuss some implications. This is a timely topic and requires only minor changes.

1. On lines 69-71, introducing the concept of suitability factor, but now defining it may introduce confusion. Either expand or remove the suitability factor percentage and simplify.

2. Optional: Mention of specific outbreaks of VBDs (chikungunya in Italy perhaps) with specific years and numbers in the introduction to underscore the importance of this work.

3. Make sure Aedes is always in italics.

4. There are no direct results from the mosquito trapping reported (outside of the abstract). Actual numbers of trapped mosquitoes and species would benefit the paper, and not only the modelled mosquito densities, especially when the importance of such trapping and surveillance is emphasized in the discussion.

5. An expansion in the discussion of previous reports of imported cases of the mentioned arboviruses would also be interesting.

6. A slight expansion in the discussion of the implications of these findings and diving deeper into prevention would also be a benefit. How effective would public information campaigns about standing water, or mosquito bite prevention in summer months be? Should insecticide campaigns be governmental, local, personal?

Reviewer #3: See attached document

PLOS authors have the option to publish the peer review history of their article (what does this mean? ). If published, this will include your full peer review and any attached files.

**Do you want your identity to be public for this peer review?** For information about this choice, including consent withdrawal, please see our Privacy Policy .

Reviewer #1: No

Reviewer #2: No

Reviewer #3: **Yes: ** Victoria Cox

**Figure resubmission:**

**Reproducibility:**



---

## [Decision Letter · Decision Letter 1]

20 Nov 2025

Dear Dr. Guzzetta,

We are pleased to inform you that your manuscript 'Assessing risks of dengue, chikungunya and Zika transmission associated to Aedes albopictus in Chania, Greece, 2017-2018' has been provisionally accepted for publication in PLOS Neglected Tropical Diseases.

Best regards,

Olaf Horstick, FFPH(UK)

Academic Editor

David Safronetz

Section Editor

Shaden Kamhawi

co-Editor-in-Chief

Paul Brindley

co-Editor-in-Chief

Reviewer's Responses to Questions

**Key Review Criteria Required for Acceptance?**

**Methods**

-Are the objectives of the study clearly articulated with a clear testable hypothesis stated?

-Is the study design appropriate to address the stated objectives?

-Is the population clearly described and appropriate for the hypothesis being tested?

-Is the sample size sufficient to ensure adequate power to address the hypothesis being tested?

-Were correct statistical analysis used to support conclusions?

-Are there concerns about ethical or regulatory requirements being met?

Reviewer #1: (No Response)

Reviewer #3: (No Response)

**Results**

-Does the analysis presented match the analysis plan?

-Are the results clearly and completely presented?

-Are the figures (Tables, Images) of sufficient quality for clarity?

Reviewer #1: (No Response)

Reviewer #3: (No Response)

**Conclusions**

-Are the conclusions supported by the data presented?

-Are the limitations of analysis clearly described?

-Do the authors discuss how these data can be helpful to advance our understanding of the topic under study?

-Is public health relevance addressed?

Reviewer #1: (No Response)

Reviewer #3: (No Response)

**Editorial and Data Presentation Modifications?**

Reviewer #1: (No Response)

Reviewer #3: (No Response)

**Summary and General Comments**

Reviewer #1: The authors have satisfactorily addressed all previous comments, and I have no further concerns.

Reviewer #3: The changes made after the first round of review have improved the clarity of the methods and greatly improved the scope of the discussion.

PLOS authors have the option to publish the peer review history of their article (what does this mean? ). If published, this will include your full peer review and any attached files.

**Do you want your identity to be public for this peer review?** For information about this choice, including consent withdrawal, please see our Privacy Policy .

Reviewer #1: No

Reviewer #3: **Yes: ** Victoria M Cox

---

## [Editor Report · Acceptance letter]

Dear Dr. Guzzetta,

We are delighted to inform you that your manuscript, "Assessing risks of dengue, chikungunya and Zika transmission associated to Aedes albopictus in Chania, Greece, 2017-2018," has been formally accepted for publication in PLOS Neglected Tropical Diseases.

Best regards,

Shaden Kamhawi

co-Editor-in-Chief

Paul Brindley

co-Editor-in-Chief
